# Type 1 Corticotropin-Releasing Factor Receptor Differentially Modulates Neurotransmitter Levels in the Nucleus Accumbens of Juvenile versus Adult Rats

**DOI:** 10.3390/ijms231810800

**Published:** 2022-09-16

**Authors:** Juan Zegers-Delgado, Alejandro Aguilera-Soza, Florencia Calderón, Harley Davidson, Daniel Verbel-Vergara, Hector E. Yarur, Javier Novoa, Camila Blanlot, Cristian P. Bastias, María Estela Andrés, Katia Gysling

**Affiliations:** Department of Cellular and Molecular Biology, Faculty of Biological Sciences, Pontificia Universidad Católica de Chile, Avenida Del Libertador Bernardo O’Higgins 340, Santiago 8331150, Chile

**Keywords:** CRF receptor 1, juvenile versus adult rats, nucleus accumbens neurotransmitter levels

## Abstract

Adversity is particularly pernicious in early life, increasing the likelihood of developing psychiatric disorders in adulthood. Juvenile and adult rats exposed to social isolation show differences in anxiety-like behaviors and significant changes in dopamine (DA) neurotransmission in the nucleus accumbens (NAc). Brain response to stress is partly mediated by the corticotropin-releasing factor (CRF) system, composed of CRF and its two main receptors, CRF-R1 and CRF-R2. In the NAc shell of adult rats, CRF induces anxiety-like behavior and changes local DA balance. However, the role of CRF receptors in the control of neurotransmission in the NAc is not fully understood, nor is it known whether there are differences between life stages. Our previous data showed that infusion of a CRF-R1 antagonist into the NAc of juvenile rats increased DA levels in response to a depolarizing stimulus and decreased basal glutamate levels. To extend this analysis, we now evaluated the effect of a CRF-R1 antagonist infusion in the NAc of adult rats. Here, we describe that the opposite occurred in the NAc of adult compared to juvenile rats. Infusion of a CRF-R1 antagonist decreased DA and increased glutamate levels in response to a depolarizing stimulus. Furthermore, basal levels of DA, glutamate, and γ-Aminobutyric acid (GABA) were similar in juvenile animals compared to adults. CRF-R1 protein levels and localization were not different in juvenile compared to adult rats. Interestingly, we observed differences in the signaling pathways of CRF-R1 in the NAc of juveniles compared to adult rats. We propose that the function of CRF-R1 receptors is differentially modulated in the NAc according to life stage.

## 1. Introduction

Adverse circumstances during childhood or adolescence can trigger neurochemical and behavioral changes that impact adulthood [1,2,3,4,5,6]. Several studies show that animals exposed to adversity in early life develop depression, cognitive disorders, anxiety, and changes in the CRF system in adulthood [4,6,7,8,9].

Stress brain response is mediated in part by the CRF system [10]. CRF is a 41-amino acid peptide synthesized in the hypothalamus and modulates the hypothalamus-pituitary-adrenal axis [11]. There are two main receptors for CRF in the brain, CRF-R1 and CRF-R2. Both are G protein-coupled receptors (GPCRs) [12,13]. CRF-R1 and CRF-R2 share almost 70% of identity; nevertheless, the N-terminals diverge, resulting in low plasma membrane presence of CRF-R2 compared to CRF-R1 [14,15,16,17,18]. CRF-R1 is present at presynaptic and postsynaptic densities [18,19]. Although CRF-R2 is mainly associated with postsynaptic density, it has also been identified in presynaptic terminals [11,20,21,22].

CRF and its receptors are widely expressed in multiple brain areas, including the ventral tegmental area (VTA), prefrontal cortex (PFC), nucleus accumbens (NAc), dorsal raphe nucleus (DRN), and others [23,24]. Several lines of evidence indicate that CRF-R1 is a critical factor in controlling behaviors such as conditioned fear, stress-induced anhedonia, anxiety-like behaviors, and others [25,26,27]. Furthermore, CRF-R1 is a critical modulator in the amygdalar response to chronic stress [28,29]. On the other hand, CRF-R2 controls basolateral amygdala-PFC circuitry [21,22].

CRF-R1 is key in controlling anxiety-like behaviors; however, both anxiogenic and anxiolytic roles have been described [30,31,32,33,34]. The NAc is a key area in motivation and goal-directed behaviors [35,36,37]. The infusion of CRF in the NAc shell induces anxiety-like behavior and changes local dopamine (DA)/acetylcholine balance [38]. Lemos and cols. showed that severe stress switches CRF action in the NAc from appetitive to aversive [39]. Furthermore, CRF induced a DA increase in the NAc, and both CRF-R1 and CRF-R2 are required for this effect [39].

Previous results of our laboratory showed that social isolation of juvenile rats increased K+-stimulated DA levels in the NAc shell assessed by in vivo microdialysis [40]. In addition, local infusion of CRF-R1 antagonist in group-housed rats increased NAc K+-stimulated DA levels, but this effect was occluded in isolated juvenile rats. On the other hand, local infusion of CRF-R1 antagonist decreased NAc glutamate basal levels in group-housed rats, and this effect was also occluded in isolated juvenile rats [40]. Altogether, our previous results suggest that neurochemical changes observed in the NAc induced by social isolation during youth are mediated by a reduced response of the CRF-R1 receptor. Since juvenile and adult rats exposed to social isolation show opposite responses in anxiety-like behaviors and significant changes in NAc DA activity [8,41,42,43], we hypothesized that the differences between juvenile and adult rats are due to differences in the control that CRF-R1 receptors exert over neurotransmission in the NAc.

This study aimed to compare the role of CRF-R1 in controlling neurotransmitter levels in the NAc between juvenile and adult rats. Our study provides evidence for opposite roles of CRF-R1 in the NAc of adults compared to juvenile rats. Data show that the infusion of a CRF-R1 antagonist into the NAc decreased DA levels and increased glutamate following a depolarizing stimulus in adult rats, opposite to what was observed in juveniles. We rule out that this is due to differences in the localization or levels of CRF-R1. We suggest that the differential function is due to different signaling downstream of CRF-R1 that would control neurotransmitter levels in NAc in an age-dependent manner.

## 2. Results

### 2.1. CRF-R1 Maintains an Inhibitory Tone upon DA and an Excitatory Tone upon Glutamate in the NAc of Juvenile Rats

To determine the role of CRF-R1 upon neurotransmitter levels in the NAc of juvenile rats, we performed in vivo microdialysis. We reproduced the effect of an intra-NAc infusion of CP-154,526 (CRF-R1 antagonist) on DA and glutamate that we previously reported [40]. Statistical comparisons between groups reveal that CP-154,526 increased DA levels in response to a depolarizing stimulus but no changes in basal levels were observed between groups (Figure 1A: Two-way ANOVA: effect of treatment, F (1, 9) = 2.019, *p* = 0.1890; effect of time, F (9, 81) = 15.25, *p* < 0.0001 and interaction, F (9, 81) = 3.491, *p* = 0.0011). Regarding glutamate, the infusion of CP-154,526 significantly decreased basal and K+-stimulated glutamate levels (Figure 1B: Two-way ANOVA: effect of treatment, F (1, 9) = 5.639, *p* = 0.0416; effect of time, F (9, 81) = 8.349, *p* < 0.0001 and interaction, F (9, 81) = 3.529, *p* = 0.0010). No changes in basal and stimulated GABA levels were observed in the presence of CP-154,526 (Figure 1C: Two-way ANOVA: effect of treatment, F (1, 9) = 0.2867, *p* = 0.6053; effect of time, F (9, 81) = 23.79, *p* < 0.0001 and interaction, F (9, 81) = 0.6012, *p* = 0.7925). These results suggest that CRF-R1 exerts opposite effects on DA and glutamate in the NAc of juvenile rats. While maintaining an inhibitory tone on DA, it stimulates glutamatergic neurotransmission.

### 2.2. CRF-R1 Maintains an Inhibitory Tone upon Glutamate and an Excitatory Tone upon DA in the NAc of Adult Rats

Using the same approach, we determined the role of CRF-R1 on neurotransmitter levels in the NAc of adult rats by directly infusing CP-154,526 during the microdialysis procedure. Statistical comparisons between groups reveal that the infusion of CP-154,526 in NAc decreased DA levels in response to a depolarizing stimulus but there was no change in basal levels (Figure 2A: Two-way ANOVA: effect of treatment, F (1, 8) = 6.258, *p* = 0.0369; effect of time, F (9, 72) = 28.51, *p* < 0.0001 and interaction, F (9, 72) = 6.401, *p* < 0.0001). This result suggests that CRF-R1 has an excitatory role in stimulated DA release. Regarding glutamate, CP-154,526 infusion significantly increased basal and stimulated glutamate levels (Figure 2B: Two-way ANOVA: effect of treatment, F (1, 8) = 16.01, *p* = 0.0039; effect of time, F (9, 72) = 6.566, *p* < 0.0001 and interaction, F (9, 72) = 6.337, *p* < 0.0001), suggesting that CRF-R1 has an inhibitory role on glutamate. No changes on basal and stimulated GABA levels were observed in the presence of CP-154,526 (Figure 2C: Two-way ANOVA: effect of treatment, F (1, 8) = 0.01630, *p* = 0.9016; effect of time, F (9, 72) = 13.39, *p* < 0.0001 and interaction, F (9, 72) = 0.07740, *p* = 0.9999). 

### 2.3. Changes in CRF-R1 Role in Juvenile versus Adult Rats

To further test whether CRF-R1 maintains opposite control over DA and glutamate in the NAc of juveniles compared to adult rats, as suggested by the above data, we statistically compared baseline neurotransmitter levels and the magnitude of the CRF-R1 antagonist infusion effect between juvenile and adult rats. Statistical comparisons between groups reveal that NAc DA (Figure 3A: non-paired Student’s *t*-test, t = 0.4519, df = 40, *p*= 0.6538), glutamate (Figure 3B: non-paired Student’s *t*-test, t = 1.802, df = 41, *p*= 0.0789), and GABA (Figure 3C: non-paired Student’s *t*-test, t = 0.6792, df = 40, *p*= 0.5009) basal levels were not different between juvenile and adult rats.

Contrary to what was observed in juvenile rats, the infusion of CP-154,526 in NAc of adult rats decreased DA levels in response to a depolarizing stimulus. Interestingly, adult rats showed a higher DA release to a depolarizing stimulus compared to juvenile rats (Figure 3D: Two-way ANOVA: effect of treatment, F (3, 25) = 3.480, *p* = 0.0308; effect of time, F (9, 225) = 42.06, *p* < 0.0001 and interaction, F (27, 225) = 4.618, *p* < 0.0001).

The infusion of CP-154,526 increased glutamate basal levels for 10 min in response to a depolarizing stimulus in the NAc of adult rats, opposed to the effect observed in juvenile rats (Figure 3E: Two-way ANOVA: effect of treatment, F (3, 25) = 9.031, *p* = 0.0003; effect of time, F (9, 225) = 9.217, *p* < 0.0001 and interaction, F (27, 225) = 3.305, *p* < 0.0001). These results suggest that CRF-R1 has an opposite role in controlling DA and glutamate neurotransmission in the NAc in juveniles compared to adult rats.

### 2.4. CRF-R1 Protein Levels in the NAc of Juvenile versus Adult Rats

To test whether the difference in the role of CRF-R1 between juvenile and adult rats could be attributed to differences in the protein level of the receptor, we performed Western blot assays. We identified three CRF-R1 isoforms in the Western blot (Figure 4A), which were present in the positive control and absent in the negative control (Appendix A). Statistical comparisons between groups reveal that CRF-R1 protein levels were not different in the NAc of juveniles compared to adult rats (Figure 4B: non-paired Student’s *t*-test t = 0.1295, df = 10, *p*= 0.8996; Figure 4C: non-paired Student’s *t*-test, t = 0.3259, df = 10, *p*= 0.7512; Figure 4D: non-paired Student’s *t*-test t = 0.9791, df = 10, *p*= 0.3506). These results suggest that the difference in the CRF-R1 role is not explained by changes in the protein levels of CRF-R1 in the NAc of juveniles compared to adult rats.

### 2.5. CRF-R1 Localization in the NAc and VTA of Juvenile versus Adult Rats

To test whether the differential role of CRF-R1 between juvenile and adult rats could be attributed to the localization of the receptor, we performed immunofluorescence in NAc and VTA. We identified specific labeling for CRF-R1, TH, and GAD67 in the soma and axon-like structures in the NAc and VTA (Appendix A). CRF-R1 labels were not different in the NAc of juveniles compared to adult rats. We observed a CRF-R1 puncta pattern in both soma and axon-like structures. Some TH-positive axon-like structures also showed a CRF-R1 puncta pattern (Figure 5A). To identify the neurochemical nature of CRF-R1-positive somas, we performed double immunofluorescence using GAD67, a marker of GABA neurons.

GAD67 labeling was not different in the NAc of juveniles compared to adult rats. Soma-like structures identified by DAPI, GAD67-positive, were also positive for CRF-R1 puncta patterns, suggesting the presence of CRF-R1 in GABA neurons. Similarly, a puncta pattern of CRF-R1 is present in some GAD67 axon-like structures (Figure 5B).

As a key part of the reward system and where dopaminergic projections to NAc were born, we evaluated the localization of CRF-R1 in the VTA of juvenile and adult rats. TH-positive soma and axon-like structures were not different in the VTA of juveniles compared to adult rats. TH-positive soma-like structures were also positive for CRF-R1 puncta patterns. Similarly, some TH axon-like structures showed a puncta pattern of CRF-R1 (Figure 5C). These results suggest that the difference in the CRF-R1 role is not explained by a difference in the localization of CRF-R1 in juveniles compared to adult rats.

### 2.6. CRF-R1 Signaling in NAc of Juvenile versus Adult Rats

Finally, we tested whether the difference in the role of CRF-R1 between juvenile and adult rats is due to differences in the signaling pathway of the receptor. To this end, we performed in vivo microdialysis co-infusing CP-154,526 with Forskolin, which induces adenylyl cyclase activity. In juvenile rats, statistical comparisons between groups reveal that Forskolin did not modify the stimulatory effect of CP-154,526 on DA in the NAc (Figure 6A: Two-way ANOVA: effect of treatment, F (2, 13) = 1.279, *p* = 0.3110; effect of time, F (9, 117) = 25.19, *p* < 0.0001 and interaction, F (18, 117) = 1.914, *p* = 0.0209). This result indicates that Gs does not mediate the inhibitory effect of CRF-R1 on stimulated DA. Regarding glutamate in juvenile rats, Forskolin reversed the inhibitory effect of CP-154,526 on basal glutamate levels but did not reverse CP-154,526 inhibition on K+-stimulated glutamate release. (Figure 6B: Two-way ANOVA: effect of treatment, F (2, 14) = 2.756, *p* = 0.0979; effect of time, F (9, 126) = 12.04, *p* < 0.0001 and interaction, F (18, 126) = 2.207, *p* = 0.0058). These results suggest that Gs at least partially mediates the excitatory effect of CRF-R1 on glutamate in juvenile rats. Furthermore, no differences in basal or stimulated GABA levels were observed with co-infusion of CP-154,526 with Forskolin compared with CP-154,526 alone. (Figure 6C: Two-way ANOVA: effect of treatment, F (2, 14) = 0.1510, *p* = 0.8612; effect of time, F (9, 126) = 34.07, *p* < 0.0001 and interaction, F (18, 126) = 0.3438, *p* = 0.9942).

The same approach was used in adult rats. Statistical comparisons between groups reveal that Forskolin restored DA release in response to a depolarizing stimulus attenuated by CP-154,526 infusion (Figure 7A: Two-way ANOVA: effect of treatment, F (2, 21) = 3.278, *p* = 0.0577; effect of time, F (9, 189) = 32.31, *p* < 0.0001 and interaction, F (18, 189) = 3.246, *p* < 0.0001). This result suggests that Gs mediates the excitatory effect of CRF-R1 upon K+-stimulated DA. Regarding glutamate, Forskolin did not modify the stimulatory effect of CP-154,526 on K+-stimulated glutamate release (Figure 6B: Two-way ANOVA: effect of treatment, F (2, 21) = 2.060, *p* = 0.1524; effect of time, F (9, 189) = 9.253, *p* < 0.0001, and interaction, F (18, 189) = 1.803, *p* = 0.0274). These results suggest that Gs does not mediate the inhibitory effect of CRF-R1 upon glutamate.

No changes on basal or stimulated GABA levels were observed by the co-infusion of CP-154,526 with Forskolin, like the infusion of CP-154,526 alone (Figure 2C: Two-way ANOVA: effect of treatment, F (2, 21) = 0.03616, *p* = 0.9645; effect of time, F (9, 189) = 19.32, *p* < 0.0001 and interaction, F (18, 189) = 0.1907, *p* > 0.9999).

## 3. Discussion

In the present study, we compared the role of CRF-R1 in the NAc at two different life stages, juvenile versus adulthood. Our data show that CRF-R1 exerts opposite functions on DA and glutamate neurotransmission in the NAc between juvenile and adult rats. We confirm our previous results in juvenile rats showing that CRF-R1 maintains an inhibitory tone on DA and a stimulatory tone on glutamate [40]. In this work, we add that in adult rats, CRF-R1 maintains an excitatory tone on DA and an inhibitory tone on glutamate. The biochemical analyses of CRF-R1 allow us to rule out that the levels or the location of the receptor explain these differences. Finally, we addressed the signaling pathways that mediate the actions of CRF-R1 and noted that the stimulatory effects on glutamate in juvenile rats and DA in adult rats are mediated by Gs, suggesting an age-dependent switch in CRF-R1 action.

### 3.1. Development of Dopaminergic and Glutamatergic Synapsis in the NAc

Several lines of evidence show that the brain’s expression of DA, glutamate, and GABA receptors changes during development [44,45,46,47]. It is an accepted idea that subcortical nuclei mature earlier than cortical ones [46]. Consistent with this evidence, we found that basal levels of DA, glutamate, and GABA were not different in the NAc, a subcortical nucleus, of juvenile compared to adult rats. Surprisingly, we found that the increase in DA levels after a depolarizing stimulus was lower in juveniles than in adult rats. This difference suggests that the system regulating DA release is not fully developed in the time window we evaluated. Supporting this idea, DA content in the striatum increases during development until adulthood [48]. Furthermore, previous studies have confirmed that the electrophysiological properties of medium spiny neurons in rat NAc mature postnatally, thus modulating the DA release differentially [49,50,51,52]. In conclusion, the intrinsic properties of the NAc change and mature during postnatal development.

### 3.2. Age-Dependent Switch of CRF-R1 Role in NAc

Our data show that CRF-R1 modulates DA and glutamate levels in the NAc of juvenile and adult rats. In juvenile rats, CRF-R1 maintains an inhibitory tone on DA and a stimulatory tone on glutamate [40]. We found that CRF-R1 is present in a puncta pattern on TH-positive axon-like structures in the NAc of juvenile rats. Similar to our results, CRF-R1 has been identified in a puncta pattern on TH-positive axon-like structures and somas of the NAc [39,53]. This evidence suggests that in the NAc, CRF-R1 is localized in DA varicosities and modulates DA neurotransmission. Further studies are needed to confirm the presence of CRF-R1 in glutamate terminals of the NAc. Nevertheless, our results suggest that CRF-R1 is also localized in glutamate terminals of the NAc due to the stimulatory tone on glutamate described here.

In adult rats, CRF-R1 maintains an excitatory tone on DA and an inhibitory tone on glutamate. Similar to our results, it was shown that CRF induces an increase in DA levels in the NAc of adult mice and rats [38,39]. Furthermore, we did not identify differences in the CRF-R1 localization between juvenile and adult rats. This evidence suggests that in the NAc of adult rats, CRF-R1 is also localized in both DA varicosities and glutamate terminals.

Though not explored in depth, NAc interneural cholinergic synapses may also play a role in the phenotypes evidenced in our study. Cholinergic interneurons modulate dopaminergic synapses in the NAc [54] through CRF-R1 signaling in adult mice [53]. Our results in adults strengthen this idea, as the antagonist of CRF-R1 reduced DA extracellular levels after a depolarizing stimulus, suggesting that CRF-R1 is essential in DA release from NAc. Furthermore, CRF induces an increase in acetylcholine levels in the NAc of adult rats [38]. We suggest that potential differences in acetylcholine levels between juvenile and adult rats could be involved in the age-dependent switch described here. Further studies are required to evaluate the role of cholinergic synapses comparing juvenile versus adult rats. 

Our data show that CRF-R1 exerts opposite functions on DA and glutamate neurotransmission in the NAc between juveniles and adult rats. Interestingly, it is not strange that some proteins exert divergent roles or expression patterns at different stages of development [55,56,57,58,59,60]. We asked whether the difference in the role of CRF-R1 could be attributed to the level or localization of the receptor; nevertheless, our results allow us to rule out this explanation. Supporting our data, it was shown that the expression of both CRF-R1 and CRF-R2 remains constant in rats through the juvenile to adult stage [60]. Although, changes in CRF-R1 have been shown in cortical structures such as the hippocampus and PFC during development [61]. Even if we did not observe differences between juvenile and adult rats regarding CRF-R1 localization, we cannot rule out the possibility that CRF-R1 levels could be different in DA varicosities or glutamate terminals at different ages. In this regard, it has been shown that quinpirole (an agonist of DA 2 receptor) increases the amplitude of evoked synaptic response and the frequency of spontaneous synaptic events in the NAc of adult rats, while in adolescent rats, it has the opposite effect. These data indicate that the DA-glutamate interactions are different between juveniles and adult animals [58,59]. Further studies are required to evaluate if CRF-R1 levels could be different in DA varicosities or glutamate terminals at different ages.

We asked whether the difference in the role of CRF-R1 could be attributed to the signaling pathway involved. It has been shown that CRF-R1 changes between Gs and Gi downstream signaling depending on agonist concentration [62]. Furthermore, the evidence pointed out that the signaling pathway involved in CRF-R1 effects could be Gq or Gs depending on the brain area or cellular model used [11,12,19,39,53,63,64,65,66,67,68]. Interestingly, homodimerization, heterodimerization, or interaction with other non-GPCR proteins could change the signaling pathway of GPCRs [18,69,70,71,72,73]. CRF-R1 is capable of homodimerization and heterodimerization with other GPCRs [18,70,72,74]. For instance, the interaction of CRF-R1 with the orexin 1 receptor, 5-hydroxytryptamine (5-HT) 2 receptor, and RAMP2 affect its downstream signaling [70,71,72]. Furthermore, evidence has pointed out that the expression of some GPCRs like DA type 1 receptors, DA type 2 receptors, 5-HT 1 receptors, and 5-HT 2 receptor changes between juvenile and adult stages [46,75,76]. We suggest that changes in the CRF-R1 homodimer/monomer balance or interaction with other receptor or non-GPCRs proteins could be involved in the age-dependent switch described here. Further studies are needed to assess whether the interaction with other proteins explains the age-dependent differences in the role of CRF-R1. 

It is important to point out that we only evaluated males in this study. Nevertheless, it has been shown that stress triggers differentially effects in male and female animals at different ages, suggesting that the role of CRF-R1 could be affected by sex [60,77]. Further studies should evaluate if sex modulates the role of CRF-R1 at different ages.

## 4. Materials and Methods

### 4.1. Animals

Juvenile male Sprague-Dawley rats, weighing 120 to 140 g and 31–34 postnatal days (PND), and young adult male Sprague-Dawley rats, weighing 280 to 320 g and 55–65 postnatal days (PNDs), were housed in pairs in cages with food and water ad-libitum. We used a total of 45 animals for this study. The colony was maintained in a temperature-controlled room (22 ± 2 °C) under a 12 h light-dark cycle with lights on at 07:00 hr. Rats were obtained from the UC CINBIOT Animal Facility of Pontificia Universidad Católica de Chile. The experimental protocols were approved by the Institutional Bioethical Committee of Pontificia Universidad Católica de Chile (N° 190211003 and 20113003). 

### 4.2. Drugs and Chemicals

CP-154,526 is a selective, non-peptide, and brain-penetrant CRF-R1 antagonist [78,79]. It was purchased from Tocris Bioscience (Cat. No. 2779) and dissolved in artificial cerebrospinal fluid (aCSF) at 10 μM. The concentration of CP- 154,526 perfused through the microdialysis probe was chosen according to our previous work [40,80]. Forskolin (adenylyl cyclase activator) was purchased from Merck (Saint Louis, MO, USA) (and dissolved in 0.1% DMSO at 30 μM.

### 4.3. In Vivo Microdialysis

Microdialysis was conducted as previously described [40,81]. Briefly, rats were anesthetized with 8% chloral hydrate (400 mg/kg, I.P.) and placed in stereotaxic apparatus. Body temperature was maintained at 37 °C, and every 60 min, a quarter of the initial dose of chloral hydrate was given to maintain the animal under deep anesthesia. It is important to state that in recent years, ethical concerns have been raised regarding the role of chloral hydrate as an anesthetic [82]. Even though chloral hydrate has been classically accepted as an anesthetic that induces unique physiological effects with minimal disruptions of NMDA and GABA transmission [83,84]. A concentric microdialysis probe, 2 mm in length (MAB 2.14.2, 35 KDa, Microbiotech) was implanted in the juvenile NAc (AP = +1.5 mm, ML = 1.3 mm, and DV = −8.5 mm, at an angle of 20°) and in the adult NAc (AP = +1.6 mm, ML = 0.9 mm and DV = −7.4 mm). The coordinates were chosen according to the stereotaxic rat atlas of Paxinos and Watson [85]. Artificial cerebrospinal fluid (aCSF) at a rate of 2 µL/min using a Harvard infusion pump (Model 22; Dover, MA, USA) was perfused through the microdialysis probe. After 90 min of the stabilization period, samples were collected in 4 µL of 0.2 N perchloric acid every 10 min. aCSF (100 μM). To stimulate neurotransmitter release, a 70 mM KCl-aCSF was perfused through the microdialysis probe for 10 min. At the end of the experiments, rats were euthanized, and the brains were quickly removed and stored in 4% paraformaldehyde for verification of probe locations (Appendix A). Quantification of DA was carried out using HPLC-electrochemical determination as previously described [80]. Briefly, 10 µL of dialysates were injected into an HPLC system (BAS, West Lafayette, IN, USA) equipped with a pump (model PM-80), an Instersil ODS-3 column (100 mm × 3.0 mm × 3 μm, G.L. science), and an amperometric detector (set at 650 mV, 0.2 nA; model LC-4C). The mobile phase, containing 0.1 M NaH2PO4, 1.0 mM 1-octanesulfonic acid, 1.2 mM EDTA, and 5% CH3CN (pH adjusted to 4.0), was pumped at a flow rate of 0.5 mL/min. Glutamate and γ-Aminobutyric acid (GABA) were detected with a fluorometric detector as previously described [40,80,81]. Briefly, 10 µL of dialysate was mixed with 10 µL of ddH2O, 4 µL of borate buffer (pH 10.8), and 4 µL of fluorogenic reagent (20 mg of ortho-phthaldehyde and 10 μL 2- Mercaptoethanol in 5 mL of absolute ethanol). A total of 20 µL of the mixture was injected into the HPLC. The mobile phase was 0.1 mM NaH2PO4 and 23.5% *v*/*v* CH3CN (pH 5.7).

### 4.4. Protein Extraction and Quantification

The left hemispheres (probe-free) of the rat brains subjected to microdialysis experiments were used for Western blots. The NAc was identified and dissected. The pieces of tissue were homogenized in 300 μL of a lysis solution (400 μL RIPA buffer (Millipore, Burlington, MA, USA), 400 μL Complete Mini protease inhibitor (Roche Diagnostics, Mannheim, Germany), 266.7 μL NaF, and 51.2 μL Na3VO4). Samples were sonicated (Kontes micro ultrasonic cell disrupter, KT50) 3 times for 10 s. Subsequently, the samples were shaken at 4 °C for 20 min. Finally, the tissues were centrifuged at 4 °C for 30 min at 15,000 RPM, and the supernatant recovered. Proteins were stored at −80 °C until quantification with the micro-BCATM Protein Assay Kit (Thermo Scientific, Pierce, Madrid, Spain) using an Epoch™ microplate spectrophotometer (BioTek^®^, Winooski, VT, USA).

### 4.5. Western Blot

A total of 40 ug of protein samples were heated in a thermal cycler (TECHNE, San Diego, CA, USA) at 37 °C for 30 min, then loaded onto a 10% SDS-PAGE gel. Electrophoresis was performed in a chamber filled with buffer (25 mM Tris-base, 192 mM Glycine, 10% SDS) using a constant voltage of 70 V for the first 30 min and then followed by a constant voltage of 100 V until the end of the proteins reached the end of the gel. Subsequently, proteins were transferred for 90 min at 300 mA while submerged in a cold buffer (25 mM Tris-base, 192 mM Glycine, 10% SDS, 20% methanol) onto a 45 µm methanol activated PVDF membrane attached to the gel via a transfer sandwich, following the methodology we previously used [86]. Once the transfer was completed, the membrane was blocked in a blocking solution (5% non-fat dry milk (Svelty), dissolved in 1 × −0.05% TBS solution of Tween 20) for 1 h with orbital shaking. Once the membrane was blocked, it was incubated overnight with the primary antibodies; mouse anti-GAPDH (1:10,000) (Millipore, MAB 374) or rabbit anti-CRF-R1 (1:1000) (Thermofisher, 720290, Waltham, MA, USA). After primary antibodies incubation, membranes were washed three times in a solution of 1 × TBS and 0.05% Tween 20 for 10 min and finally incubated with the peroxidase-conjugated secondary antibodies: α-Rabbit (1:4000) (Jackson Immunoresearch, AB_10015282), α-Mouse (1:2500) (Jackson Immunoresearch, AB_2340770). After incubation with secondary antibodies, the membranes were washed four times, for 10 min each, with the solution described before. Immunodetection was performed using SuperSignal^®^ West Pico Chemiluminiscent Substrate (Thermo Scientific Inc., Pierce, Madrid, Spain). The detection for each protein was performed separately from the other. 

### 4.6. Immunofluorescence

Immunofluorescence was performed as previously described [86]. Briefly, animals were transcardially perfused with 4% paraformaldehyde. Brains were removed and postfixed in 4% paraformaldehyde, then transferred to a 20% sucrose solution. Brains were cut into 30 μm slices using a Leica CM 1510 cryostat. Slices were washed three times with PBS 1 × solution, ten minutes each, then treated with 1% NaBH4 for fifteen minutes and washed again with PBS. Slices were incubated with 0.3% Triton X−100 (Winkler) dissolved in PBS 1 ×, then washed three times with a 0.05% Triton X−100 solution. Blocking was performed by incubating slices for two hours in a PBS 1 × solution with 10% normal donkey serum (Merck) and 0.05% Triton X−100. Primary antibodies—mouse anti-tyrosine hydroxylase (TH) (1:1000, Sigma), rabbit anti-CRF-R1 (4:1000, Thermo Fisher, Waltham, MA, USA), mouse anti-glutamate decarboxylase (GAD)67 (1:1000, Merck) were incubated at 4 °C overnight, dissolved in a solution made of PBS 1 × with 5% normal donkey serum and 0.05% Triton X−100. The next day, slices were washed three times with 1% normal donkey serum, 0.05% Triton X−100, PBS 1 ×, ten minutes each, then incubated at room temperature in a dark chamber with secondary antibodies—Alexa Fluor goat anti-mouse 568 (1:2000, Thermo Fisher), Alexa Fluor donkey anti-rabbit 488 (1:2000, Thermo Fisher)-for two hours. Finally, slices were washed two times with 1% normal donkey serum, 0.05% Triton X−100, PBS 1 × solution, and two times with PBS 1 ×, ten minutes each. Slices were mounted on glass slides using a mounting medium with DAPI (DAPI Fluoromount G™, Electron Microscopy Science, Mannheim, Germany). Image acquisition was made using a Zeiss LSM 880 confocal microscope. In order to increase the visibility of structures and puncta patterns, every image was brightness modified, and the background was subtracted as described in Appendix A. Furthermore, this figure shows every photo available before the adjustment (Appendix A). The details and codes of every antibody are described in Appendix A.

### 4.7. Statistical Analysis

Statistical differences were analyzed, and graphs were generated in Prism 6 software (GraphPad Software, San Diego, CA, USA). No randomization was performed to allocate subjects in this study. All data were normal, according to the D’Agostino and Pearson normality test performed. The number of rats used is indicated in the figure legend of each experiment. The statistical analysis used varied according to the data evaluated. The comparison of DA, glutamate, and GABA basal levels between juvenile and adult rats was analyzed with a non-paired, two-sided Student’s *t*-test. Comparing CRF-R1 protein levels between juvenile and adult rats were analyzed with a non-paired, two-sided Student’s *t*-test. For comparing the CP-154,526 effect in juvenile and adult rats, a two-way ANOVA followed by Bonferroni post hoc test was performed in all microdialysis experiments. An alpha level of *p* < 0.05 was considered statistically significant. All data were presented as mean ± SEM.

## 5. Conclusions

The main finding of our work is that CRF, through its receptor CRF-R1, has opposite roles between juveniles and adults in controlling DA and glutamate neurotransmission. Further, the stimulatory effects on glutamate in juveniles and DA in adults are mediated by Gs, suggesting an age-dependent change in downstream CRF-R1 signaling. Further studies will be needed to analyze the mechanism that could explain the switch of CRF-R1 between juvenile and adult rats. It is important to note that CRF-R2 also plays a key role in stress response regulation, and future studies should address the role of CRF-R2 at NAc synapses during development.

## Figures and Tables

**Figure 1 ijms-23-10800-f001:**
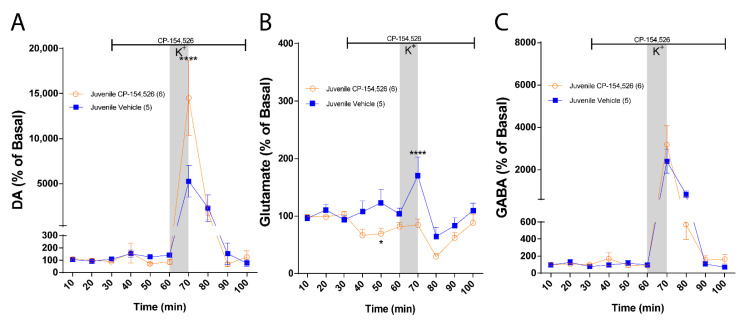
CRF-R1 modulates neurotransmitter levels in NAc of juvenile rats. Measurement of NAc extracellular neurotransmitter levels using in vivo microdialysis, the vertical gray bar indicates the time of NAc local infusion of 70 mM K+-aCSF, and the horizontal black lines indicate the time of intra-NAc infusion of the antagonist or vehicle. (**A**) NAc DA levels in the presence of aCSF (blue; *n* = 5) and 10 µM of CP154,526 (orange; *n*
*=* 6). DA levels were normalized to the average of the first 3 values for each condition and expressed as a percentage. Two-way ANOVA revealed no effect of treatment (F (1, 9) = 2.019, *p* = 0.1890), a main effect of time (F (9, 81) = 15.25, *p* < 0.0001), and treatment × time interaction (F (9, 81) = 3.491, *p* = 0.0011; **** *p* < 0.0001, CP154,526 vs. aCSF, Bonferroni post hoc test). (**B**) NAc glutamate levels in the presence of aCSF (blue; *n* = 5) and 10 µM of CP154,526 (orange; *n**=* 6). Glutamate levels were normalized to the average of the first 3 values for each condition and expressed as percentage. Two-way ANOVA revealed a main effect of treatment (F (1, 9) = 5.639, *p* = 0.0416), a main effect of time (F (9, 81) = 8.349, *p* < 0.0001), and treatment × time interaction (F (9, 81) = 3.529, *p* = 0.0010; * *p* = 0.0394, **** *p* < 0.0001, CP154,526 vs. aCSF, Bonferroni post hoc test). (**C**) NAc GABA levels in the presence of aCSF (blue; *n* = 5) and 10 µM of CP154,526 (orange; *n*
*=* 6). GABA levels were normalized to the average of the first 3 values for each condition and expressed as percentage. Two-way ANOVA revealed no effect of treatment (F (1, 9) = 0.2867, *p* = 0.6053), a main effect of time (F (9, 81) = 23.79, *p* < 0.0001), and no treatment × time interaction (F (9, 81) = 0.6012, *p* = 0.7925; n.s. *p* > 0.9999, CP154,526 vs. aCSF, Bonferroni post hoc test).

**Figure 2 ijms-23-10800-f002:**
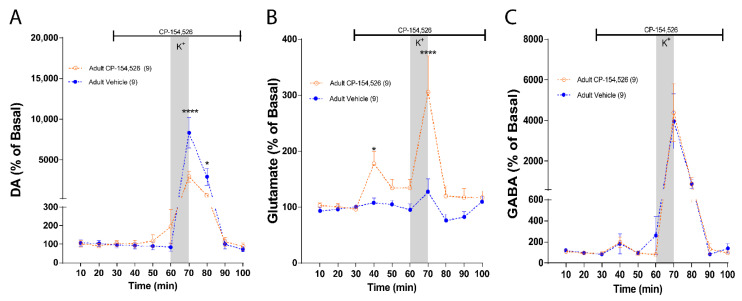
CRF-R1 modulates neurotransmitter levels in NAc of adult rats. Measurement of NAc extracellular neurotransmitter levels using in vivo microdialysis, the vertical gray bars indicate the time of NAc local infusion of 70 mM K+-aCSF, and the horizontal black lines indicate the time of intra-NAc infusion of the antagonist or vehicle. (**A**) NAc DA levels in the presence of aCSF (blue; *n*
*=* 9) and 10 µM of CP154,526 (orange; *n*
*=* 9). DA levels were normalized to the average of the first 3 values for each condition and expressed as percentage. Two-way ANOVA revealed a main effect of treatment (F (1, 8) = 6.258, *p* = 0.0369), a main effect of time (F (9, 72) = 28.51, *p* < 0.0001), and treatment × time interaction (F (9, 72) = 6.401, *p* < 0.0001; **** *p* < 0.0001, * *p* =0.0210, CP154,526 vs. aCSF, Bonferroni post hoc test). (**B**) NAc glutamate levels in the presence of aCSF (*n*
*=* 9) and 10 µM of CP154,526 (*n =* 9). Glutamate levels were normalized to the average of the first 3 values for each condition and expressed percentage. Two-way ANOVA revealed a main effect of treatment (F (1, 8) = 16.01, *p* = 0.0039), a main effect of time (F (9, 72) = 6.566, *p* < 0.0001), and treatment × time interaction (F (9, 72) = 6.337, *p* < 0.0001; * *p* = 0.0140, **** *p* < 0.0001, CP154,526 vs. aCSF, Bonferroni post hoc test). (**C**) NAc GABA levels in the presence of aCSF (*n =* 9) and 10 µM of CP154,526 (*n =* 9). GABA levels were normalized to the average of the first 3 values for each condition and expressed as percentage. Two-way ANOVA revealed no effect of treatment (F (1, 8) = 0.01630, *p* = 0.9016), a main effect of time (F (9, 72) = 13.39, *p* < 0.0001), and no treatment × time interaction (F (9, 72) = 0.07740, *p* = 0.9999; n.s. *p* > 0.9999, CP154,526 vs. aCSF, Bonferroni post hoc test).

**Figure 3 ijms-23-10800-f003:**
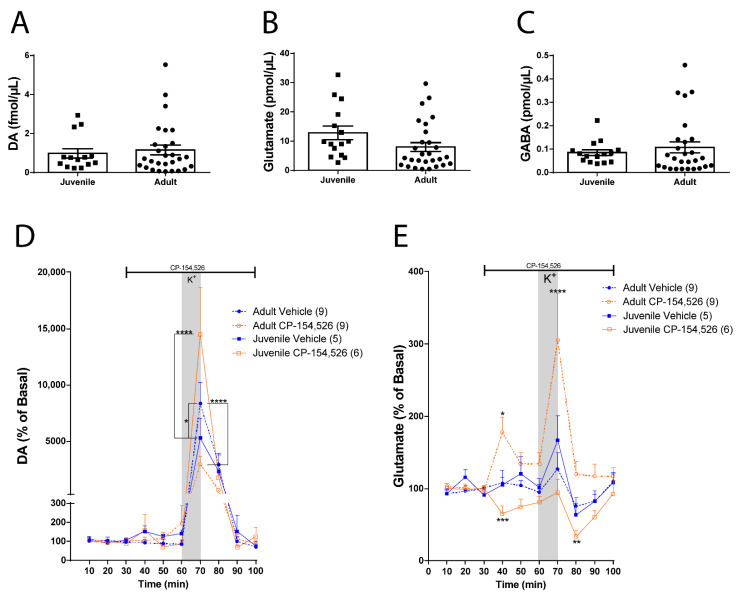
Changes in CRF-R1 role in juvenile versus adult rats. Comparison between juvenile versus adult rats. Measurement of NAc extracellular neurotransmitter basal levels and CRF-R1 role using in vivo microdialysis. (**A**) The average of the first 3 values for each of NAc DA levels was compared between juvenile (*n* = 14) and adult rats (*n* = 28). Non-paired Student’s *t*-test revealed that DA basal levels were not different in juvenile animals compared to adults (t = 0.4519, df = 40, *p*= 0.6538). (**B**) The first 3 values for each NAc glutamate level were averaged and compared between juvenile (*n* = 15) and adult rats (*n* = 28). Non-paired Student’s *t*-test revealed that basal glutamate levels were not different in juvenile animals compared to adults (t = 1.802, df = 41, *p*= 0.0789). (**C**) The first 3 values for each NAc GABA level were averaged and compared between juvenile (*n* = 15) and adult rats (*n* = 28). Non-paired Student’s *t*-test revealed that GABA basal levels were not different in juvenile animals compared to adults (t = 0.6792, df = 40, *p*= 0.5009). Measurement of NAc extracellular neurotransmitter levels using in vivo microdialysis, the vertical gray bars indicate the time of NAc local infusion of 70 mM K+-aCSF, and the horizontal black lines indicate the time of intra-NAc infusion of the antagonist or vehicle. (**D**) Adult versus juvenile NAc DA levels in the presence of aCSF (Juvenile, continued blue line, *n* = 5; Adult, blue dotted line, *n*
*=* 9) and 10 µM of CP154,526 (Juvenile, orange continued line, *n*= 6; Adult, orange dotted line, *n*
*=* 9). DA levels were normalized to the average of the first 3 values for each condition and expressed as percentage. Two-way ANOVA revealed a main effect of treatment (F (3, 25) = 3.480, *p* = 0.0308), a main effect of time (F (9, 225) = 42.06, *p* < 0.0001), and treatment × time interaction (F (27, 225) = 4.618, *p* < 0.0001; **** *p* < 0.0001, * *p* = 0.0360, Bonferroni post hoc test). (**E**) Adult vs. juvenile NAc glutamate levels in the presence of aCSF (Juvenile, blue continued line, *n*= 5; Adult, blue dotted line, *n*
*=* 9) and 10 µM of CP154,526 (Juvenile, orange continued line, *n*= 6; Adult, orange dotted line, *n*
*=* 9). Glutamate levels were normalized to the average of the first 3 values for each condition and expressed as percentage. Two-way ANOVA revealed a main effect of treatment (F (3, 25) = 9.031, *p* = 0.0003), a main effect of time (F (9, 225) = 9.217, *p* < 0.0001), and treatment × time interaction (F (27, 225) = 3.305, *p* < 0.0001; * *p* = 0.0160, *** *p* = 0.0001, **** *p* < 0.0001, ** *p* = 0.0057, Bonferroni post hoc test).

**Figure 4 ijms-23-10800-f004:**
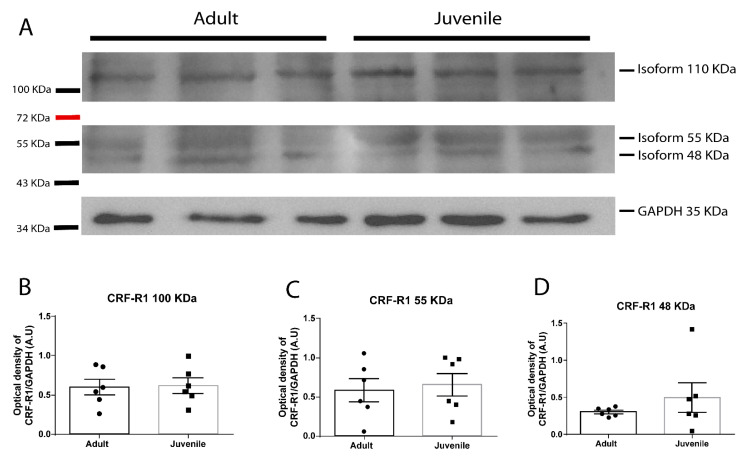
CRF-R1 protein levels in juvenile versus adult rats. Comparison of CRF-R1 protein levels between juvenile and adult rats using Western blot. (**A**) CRF-R1 predicted band size is 48 kDa. We identified three isoforms of CRF-R1 as the two bands below 55 kDa and the band upon 100 KDa. The band above 35 kDa corresponds to GAPDH. (**B**) Quantification of the optical densities of the 100 KDa CRF-R1 band normalized to the optical densities of the GAPDH band. Non-paired Student’s *t*-test revealed that 100 KDa CRF-R1 levels were not different in juvenile animals compared to adults (t = 0.1295, df = 10, *p* = 0.8996). (**C**) Quantification of the optical densities of the 55 KDa CRF-R1 band normalized to the optical densities of the GAPDH band. Non-paired Student’s *t*-test revealed that 55 KDa CRF-R1 levels were not different in juvenile animals compared to adults (t = 0.3259, df = 10, *p* = 0.7512). (**D**) Quantification of the optical densities of the 48 KDa CRF-R1 band normalized to the optical densities of the GAPDH band. Non-paired Student’s *t*-test revealed that 48 KDa CRF-R1 levels were not different in juvenile animals compared to adults (t = 0.9791, df = 10, *p* = 0.3506).

**Figure 5 ijms-23-10800-f005:**
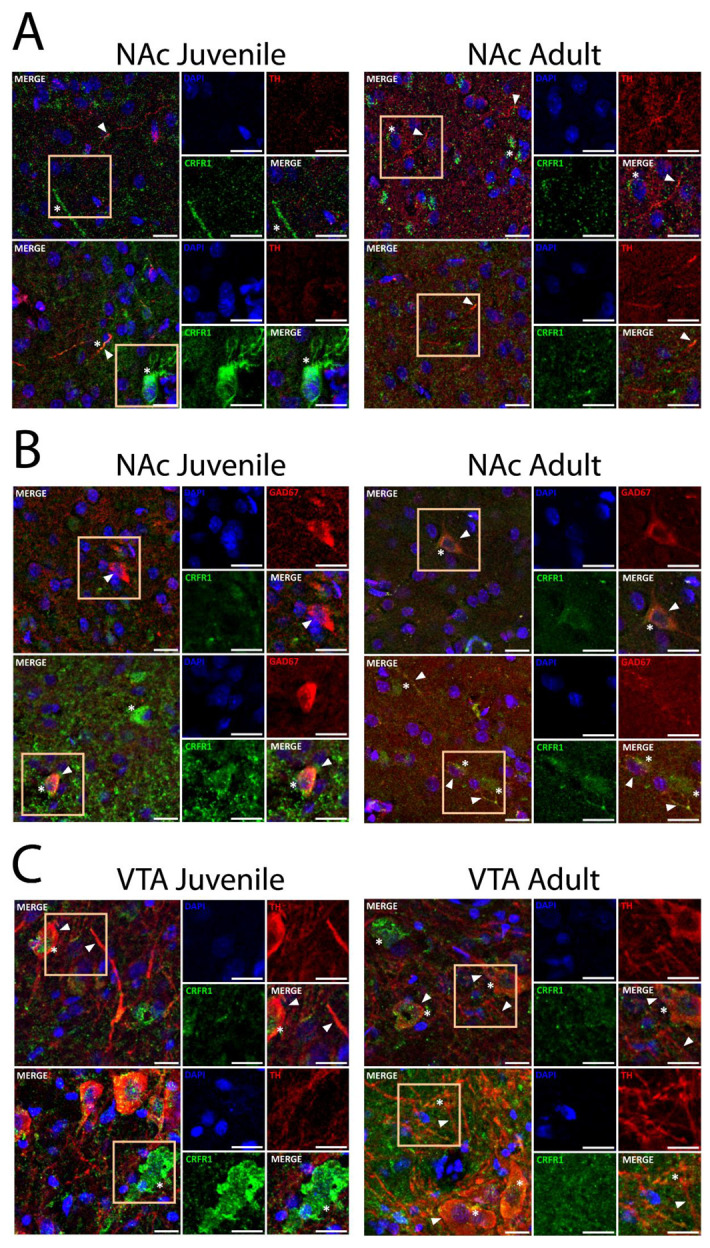
CRF-R1 localization in juvenile versus adult rats. Comparison of CRF-R1 localization between juvenile (left) and adult (right) rats using immunofluorescence. The right panel is a 1.3 × zoom of the selected region on the left, showing every separate channel and its respective merge image. The scale bar represents 15 μm in every image. (**A**) The figure shows patterns of the label for CRF-R1 and TH in the NAc of juvenile and adult rats. Arrowheads show TH-positive axons, and asterisks show puncta pattern distribution of CRF-R1. (**B**) The figure shows patterns of the label for CRF-R1 and GAD67 in the NAc of juvenile and adult rats. Arrowheads show GAD67-positive somas and axons, and asterisks show puncta pattern distribution of CRF-R1. (**C**) The figure shows patterns of the label for CRF-R1 and TH in the VTA of juvenile and adult rats. Arrowheads show TH-positive somas and axons, and asterisks show puncta pattern distribution of CRF-R1.

**Figure 6 ijms-23-10800-f006:**
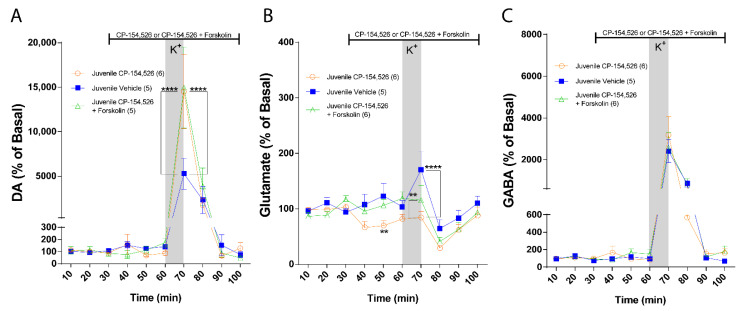
CRF-R1 signaling pathway in NAc of juvenile rats. Measurement of NAc extracellular neurotransmitter levels using in vivo microdialysis, the vertical gray bar indicates the time of NAc local infusion of 70 mM K+-aCSF, and the horizontal black lines indicate the time of intra-NAc infusion of the antagonist, vehicle, or co-infusion of antagonist with Forskolin. (**A**) NAc DA levels in the presence of aCSF (blue; *n* = 5), 10 µM of CP154,526 (orange; *n* = 6), and 10 µM of CP154,526 + 50 µM of Forskolin (green; *n* = 5). DA levels were normalized to the average of the first 3 values for each condition and expressed as percentage. Two-way ANOVA revealed no effect of treatment (F (2, 13) = 1.279, *p* = 0.3110), a main effect of time (F (9, 117) = 25.19, *p* < 0.0001), and treatment × time interaction (F (18, 117) = 1.914, *p* = 0.0209; CP154,526 vs. aCSF, **** *p*< 0.0001, CP154,526 + Forskolin vs. aCSF, **** *p*< 0.0001, Bonferroni post hoc test). (**B**) NAc glutamate levels in the presence of aCSF (blue; *n* = 5), 10 µM of CP154,526 (orange; *n* = 6) and 10 µM of CP154,526 + 50 µM of Forskolin (green; *n* = 6). Glutamate levels were normalized to the average of the first 3 values for each condition and expressed as percentage. Two-way ANOVA revealed no effect of treatment (F (2, 14) = 2.756, *p* = 0.0979), a main effect of time (F (9, 126) = 12.04, *p* < 0.0001), and treatment × time interaction (F (18, 126) = 2.207, *p* = 0.0058; CP154,526 vs. aCSF, ** *p*= 0.0098, **** *p*< 0.0001, CP154,526 + Forskolin vs. aCSF, ** *p*= 0.0079, Bonferroni post hoc test). (**C**) NAc GABA levels in the presence of aCSF (blue; *n* = 5), 10 µM of CP154,526 (orange; *n* = 6) and 10 µM of CP154,526 + 50 µM of Forskolin (green; *n* = 6). GABA levels were normalized to the average of the first 3 values for each condition and expressed as percentage. Two-way ANOVA revealed no effect of treatment (F (2, 14) = 0.1510, *p* = 0.8612), a main effect of time (F (9, 126) = 34.07, *p* < 0.0001), and no treatment × time interaction (F (18, 126) = 0.3438, *p* = 0.9942; CP154,526 vs. aCSF, n.s. *p* > 0.9999, CP154,526 + Forskolin vs. aCSF, n.s. *p* > 0.9999, Bonferroni post hoc test).

**Figure 7 ijms-23-10800-f007:**
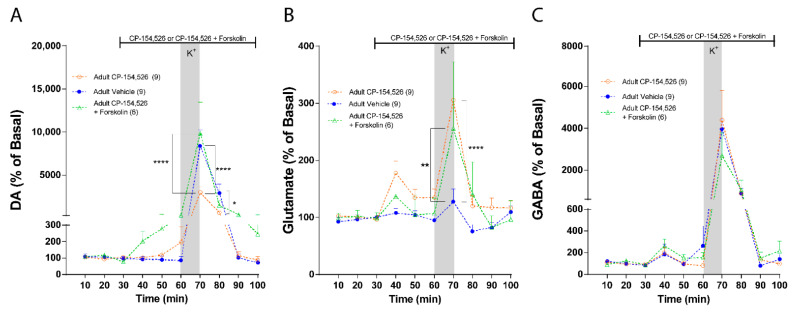
CRF-R1 signaling pathway in NAc of adult rats. Measurement of NAc extracellular neurotransmitter levels using in vivo microdialysis, the vertical gray bars indicate the time of NAc local infusion of 70 mM K+-aCSF, and the horizontal black lines indicate the time of intra-NAc infusion of the antagonist, vehicle, or co-infusion of antagonist with Forskolin. (**A**) NAc DA levels in the presence of aCSF (blue dotted line; *n*
*=* 9), 10 µM of CP154,526 (orange dotted line; *n*
*=* 9) and 10 µM of CP154,526 + 50 µM of Forskolin (green dotted line; *n* = 6). DA levels were normalized to the average of the first 3 values for each condition and expressed as percentage. Two-way ANOVA revealed no effect of treatment (F (2, 21) = 3.278, *p* = 0.0577), a main effect of time (F (9, 189) = 32.31, *p* < 0.0001), and treatment × time interaction (F (18, 189) = 3.246, *p* < 0.0001; CP154,526 vs. aCSF, **** *p* < 0.0001, * *p* = 0.0210, CP154,526 + Forskolin vs. aCSF, n.s. *p* > 0.9999, Bonferroni post hoc test). (**B**) NAc glutamate levels in the presence of aCSF (blue dotted line; *n*
*=* 9), 10 µM of CP154,526 (orange dotted line; *n*
*=* 9) and 10 µM of CP154,526 + 50 µM of Forskolin (green dotted line; *n* = 6). Glutamate levels were normalized to the average of the first 3 values for each condition and expressed as percentage. Two-way ANOVA revealed no effect of treatment (F (2, 21) = 2.060, *p* = 0.1524), a main effect of time (F (9, 189) = 9.253, *p* < 0.0001), and treatment × time interaction (F (18, 189) = 1.803, *p* = 0.0274; CP154,526 vs. aCSF, **** *p* < 0.0001, CP154,526 + Forskolin vs. aCSF, ** *p*= 0.0025, Bonferroni post hoc test). (**C**) NAc GABA levels in the presence of aCSF (blue dotted line; *n*
*=* 9), 10 µM of CP154,526 (orange dotted line; *n*
*=* 9) and 10 µM of CP154,526 + 50 µM of Forskolin (green dotted line; *n* = 6). GABA levels were normalized to the average of the first 3 values for each condition and expressed as percentage. Two-way ANOVA revealed no effect of treatment (F (2, 21) = 0.03616, *p* = 0.9645), a main effect of time (F (9, 189) = 19.32, *p* < 0.0001), and no treatment × time interaction (F (18, 189) = 0.1907, *p* > 0.9999; CP154,526 vs. aCSF, n.s. *p* > 0.9999, CP154,526 + Forskolin vs. aCSF, n.s. *p* > 0.9999, Bonferroni post hoc test).

## Data Availability

Data will be made available upon request. Please contact the corresponding authors.

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
