# Peer review of "Type 1 Corticotropin-Releasing Factor Receptor Differentially Modulates Neurotransmitter Levels in the Nucleus Accumbens of Juvenile versus Adult Rats"

_ijms, 2022, doi:10.3390/ijms231810800_

Round 1

Reviewer 1 Report

In this paper, Zegers-delgado et al attempt to show the effect of an infusion of a CRF1-R1 receptor antagonist into the NAc on the release of neurotransmitters (DA, Glu, GABA) in young adult and adolescent males. This measurement is performed in anesthetized animals by microanalysis followed by HPLC analysis. The aim of this study is to better understand the role of CRF-R1 in the NAc on the stress response.

This article is not acceptable for publication as unethical:

The anesthesia used (Chloral Hydrate) in rats is not tolerated by the majority of ethics committees worldwide. For several reasons:

1) Chloral Hydrate causes "hypnosis" and not anesthesia.

2) Chloral hydrate does not cause analgesia and induces respiratory depression

3) Chloral hydrate is extremely irritating and cannot be used in ip

4) It is associated with gastric ulcers and peritonitis in rats

No ethics committee would tolerate the use of this anesthetic for a long stereotactic operation. There is no "tenable" scientific justification for using chloral hydrate in this experiment.

Major points

1)-The study is only on males and needs to be done on females especially at this time of hormonal upheaval.

2) The article is incomprehensible, many of the references are not even in the reference list, making it difficult to read: Example in the first paragraph alone (Sutanto and de Kloet, 1994; Robbins et al., 1996; Blanchard et al., 1998; Palanza, 2001; Vargas et al., 2016; Ladd et al., 1996)...

3) The same data are presented several times in different figures (Figure 1 juveniles and Figure 2 adults are repeated in Figure 3 but also in Figure 6 and 7).

4) The observations are quite interesting: injection of the CRF-1 antagonist increases Glut levels and decreases DA levels in response to a depolarizing stimulus (KCl injection). But it would be even more interesting to functionally (electrophysiologically) swallow the effect of the antagonist on NAc MSN neurons to verify the functional impact. It would be interesting to evaluate the effect in-vivo during a behavior.

5) Many other problems: The number of animals is not always the same (n=3 for the vehicle group in figure 1 or n=5)

To conclude this study is not very informative, ethically unacceptable and poorly presented (redundancy of information, extremely poorly done references...)

Author Response

Regarding to the ethical concerns related to the use of chloral hydrate in our experiments:

  • Even though chloral hydrate could induce gastric ulcers after 6 hours i.p. in rats, it should be noted that the use of chloral hydrate in our protocol is limited to non-survival surgical procedures and limited by 4 hours of total procedure as much.
  • The duration of anesthesia induced by chloral hydrate is linearly related to dose (Field et al., 1993; Silverman & Muir, 1993).
  • Chloral hydrate still has an important role as an anesthetic agent in biomedical research due to its unique physiologic effects with minimal disruptions of NMDA and GABA transmission (Caballero et al., 2014; Flores-Barrera et al., 2014).

Major point:

1) We totally agree that analyzing the response of females in these 2 developmental periods, juveniles vs. adults, may be interesting for future studies. But we do not see it as an obligation that limits the publication of this article. This may be perfectly addressable in future research.

2) Corrected in the new version of the manuscript

3) The way in which the figures were arranged is to facilitate the reader's understanding. Furthermore, in all figures specific statistical analyses were conducted to evaluate differences within groups. We could condense all the information into 3 figures, but it seems to us that it would be incomprehensible. In addition, in the first figure we did not only replicate data from our previous studies, but also show the antagonist effect on GABA. From our point of view, including the forskolin condition in these figures (1 and 2) would make it more difficult to understand the figure. 

4) We agree that an electrophysiological and behavioral approach would reinforce the results shown, but the present results are novel by themselves and they should motivate further research.

5) Corrected in the new version of the manuscript. 

Reviewer 2 Report

In general, the concept of this work is really interesting. But there are some points of criticism which need to be clarified before the publication

Line 43 – there is no consequence in abbreviations of the brain anatomical structures. Accordingly, nucleus accumbens should be abbreviated to (NAc) and written in small (not capital) letters. Also dorsal raphe nucleus should be abbreviated to DRN but not DN.

Line 47 – please explain what DA stands for. Admittedly, this acronym is expanded in line 62, but used for the first time in line 47.

Figure 5 – all immunofluorescent images are too small to make any judgement. Please also note that some of scale bars are barely seen. This must be corrected.

Line 195 – The term “expression” should be used in relation to genes only.

Line 222 – please explain what TH and GAD67 stand for.

Line 506 - The major problem of this study is that the authors did not test properly the specificities of antibodies they used. The preadsoprtion tests or any other negative/positive controls are necessary.

Line 515 –Please provide manufacturer’s codes for antisera used.

Line 524 – 0.05% instead of 0,05%

Author Response

All minor points regarding abbreviation issues were corrected

We have chosen new photos and zoom them in order to make them bigger.

We added a table in which every antibody is described and if the manufacturer performed the pre-adsorption test. For primary antibodies no pre-adsorption test was available, but we also performed an extra experiment in order to validate the CRF-R1 antibody, using Hek293T cells as it is shown in Figure S5. Negative and positive controls for each antibody are available in Figure S3. We agree that further validation could be required using a K.O of CRF-R1 murine model but we are not able to perform this kind of experiments in this moment.

Reviewer 3 Report

The manuscript entitled ” Type 1 corticotropin-releasing factor receptor differentially modulates neurotransmitter levels in Nucleus Accumbens of juvenile versus adult rats” by Juan Zegers-Delgado and collaborators is focusing on the role of CRF-R1 in Nac of adults vs. juvenile Sprague-Dawley rats. The topic of the manuscript is relevant and very interesting. Some concerns are listed below:

The Introduction section seems rather short and not very informative. Moreover, almost half of the section is in fact focused on a previous study of the authors. Also, and the aim of the study is not clearly defined.

The authors mention that for the action of CRF, both CRF receptors are required (lines 48-49). Then, why the study is focused only on CRF-R1? What is the role of CRF-R2? Please add more information about the two receptors in the Introduction section.

Add information regarding the CRF-R1 antagonist that has been used.

For GABA (line 22) and GPCRs (line 41), please add the in extenso terms.

For glutamate, two abbreviations are mentioned: Glut (line 18) and GLU (22). Please check. Also, despite the fact that the authors provide an abbreviation for glutamate, they do not use it throughout the manuscript. Please use the abbreviation or remove it from the manuscript.

In the case of dopamine abbreviation (DA), the authors use it throughout the introduction section, but explain the it at line 62. Please check and add the abbreviation when the term has been used for the first time in a section.

For Figures 1, 2, 3, 6, and 7, the Y axis titles appear to have different dimensions for A, B, and C. Please check and revise.

For all Figures, I assume that the number in the parenthesis (e.g. Figure 1. A. CP-154,526 (6)) represent the number of animals per group. If that is true, please check all the Figures and their legends. There are some inconsistencies.

For Figure legends, the ANOVA analysis is presented as 2-way ANOVA analysis or two-way ANOVA analysis. Please, use only one variant throughout the manuscript.

For Figure 1 legend, for A Bonferoni post-hoc test has been used, for B Fisher’s LSD post-hoc has been used, and for C no post-hoc analysis has been mentioned. Please check if there are just some inconsistencies or explain why two types post-hoc analysis were applied. Same is true for other Figure legends. Please check.

Figure 5 presented in the current form are is very difficult to follow. For each picture please add an understandable scale bar and the name of the marker (e.g. DAPI,  CRFR1 etc. and Merged). Please change a, b, c, d with 1, 2, 3, 4 or other system as the authors prefer. For the zoomed pictures, please add the zoom magnification. Also, the resolution of the pictures is very low and pictures appear very fuzzy. Please, enhance the picture resolution. Moreover, the authors mention the presence of arrows and asterisks on c, but actually they appear to be on b and d. Please check and revise.

In the Materials and methods, Animals section, please add the total number of animals that has been used.   

In the Materials and methods, Protein extraction and quantification section, please add the type of sonicator that has been used.

In the Materials and methods, the Western Blot section seems to be incomplete. Please add more details about the protocol that has been used.

In the Materials and methods, the Statistical analysis section seems to be incomplete. Please add all the applied statistical analysis.

The Conclusion section includes only general information. Please improve the section.

For References sections, a variety of styles was used. Please, select just one style and use it for all references. Moreover, papers as Sutano and de Kloet, 1994; Robbins et al., 1996; Palanza, 2001; Ladd et al., 1996; etc. from Introduction section do not appear in the list of references. On the other hand, papers such as Bassareo et al., 2002, Bledsoe et al., 2011, Di Chiara, 2002; etc., appear in the references list, but not in the manuscript. Please, revise all the references.

Author Response

All points were corrected.

We changed the introduction in order to make it more straightforward and informative.

We only focused in CRF-R1 because our previous studies, but we agree that it is neccesary to evaluate CRF-R2 in future studies. We added a conclusion regarding this point.

All minor points regarding abbreviation issues were corrected.

We added information regarding CRF-R1 antagonist used.

All axis titles were modified to make them equal between figures.

All inconsistencies in the figure legends were checked and corrected.

We have chosen new photos and zoom them in order to make them bigger.

The issues regarding material and methods section were corrected.

All references were checked and corrected.

Round 2

Reviewer 1 Report

Concerning the ethical point: You are not allowed to say that "every 60  minutes, a quarter of the initial dose of chloral hydrate was given to maintain the animal under deep anesthesia to avoid suffering" choral hydrate is a poor analgesia so your animals were suffering during the surgery... At least, apply some analgesia during the experiment (ketoprofen or buprenorphine). In 2022 it is not acceptable anymore. At least address this ethical issue in the method...

Concerning the female: you didn't discuss this point in the discussion? In research we are required to address gender in our studies, if not at least you had to bring it up in the discussion.

For the other concerns the present form is better...

Author Response

Regarding the ethical concerns:

We have addressed the ethical issue in the methods section.

Regarding the sex differences:

We incorporate a brief commentary in the discussion regarding the possible sex difference in our study context.

Reviewer 3 Report

I thank the authors for addressing each of the raised concerns. 

Author Response

Thank you!